# A Background to a Methodology for Transport Mobile Applications Designing with a Focus on EU Interoperability

**Pavla Šmídová *** and **Stanislav Machalík**

Department of Information Technology in Transport, University of Pardubice, Studentská 95,
532 10 Pardubice, Czech Republic
* Correspondence: smidova.pavla@gmail.com; Tel.: +420-776-292-200

**Abstract:** European rail transport policy aims to create a single railway area. European Union (EU) issues, for this purpose, sets of regulations and directives. Individual countries are involved in its implementation according to their capabilities. The aim of this article is to examine the current state of interoperability in the EU, especially in the Czech Republic, and to compare selected mobile applications regarding the possibility of unifying their design of mobile applications. A systematic literature research review methodology was used. The sources of information were mainly the scientific database Web of Science and the database of EU documents EUR-Lex. When examining one of the main parameters—the use of mobile networks—it was found that all of the selected applications use public mobile networks, none of them uses the Global System for Mobile Communications–Railway (GSM–R) network. This finding confirmed the theoretical assumption that this network is not currently ready for more or less massive operation of mobile applications. The possibility of using the next generation railway mobile network is open for discussion as one of the next steps to ensure interoperability in telematics applications.

**Keywords:** railways; European Union; interoperability; mobile applications; GSM–R; FRMCS





## 1. Introduction

There is actually a push for standardization in many areas in the EU and rail transport is one of the most addressed topics.

At the same time, with the meteoric rise of information and mobile technologies, mobile applications are becoming more easily available both for use and development. There are many approaches to their development. Is it possible to combine the specifics of rail transport with such a rapidly developing area as information technology in the field of mobile applications, thus benefiting not only the developer but also the user? The resolution of these issues is very important in view of the EU's move towards interoperability and the integration of other systems into it.

In the existing literature, studies and analyses of the current situation, the following state of affairs was found: it is clear from EU documents and reports that a single European Rail Traffic Management System (ERTMS) has been under development for some time. Many studies have been carried out within EU projects so far, resulting in recommendations, directives, regulations, etc., in various forms of specification and impact on EU countries. Their implementation has been further monitored and published in reports. Particularly, precise specifications exist for telematics applications, GSM–R is specified in detail and the FRMCS specification is being worked on. This future change is closely monitored and discussed in studies, as it will significantly affect rail transport in EU countries. However, more extensive research into the possibility of a unified design of mobile applications with a focus on rail transport has not yet been carried out. The methodology for the design of mobile applications aims to fill this gap by combining the possibility of unifying rail transport and IT in the form of mobile applications.

As this is a large topic, the work is divided into several subunits. This paper focuses specifically on the research of current EU measures and one of the main parameters of mobile applications—connection to the mobile data network. From the results of the research, it will be possible to conclude whether the possibility of greater unification is worth considering or not.

The study of this paper aimed to answer the following questions:

What EU interoperability regulations and recommendations should be included in the methodology to make truly useful guidelines for mobile application developers and users? What is currently the real situation for the selected parameter—mobile networks in the EU—with a more detailed look at the Czech Republic and its neighbours Slovakia, Poland and Hungary? Is it possible to propose unified recommendations for mobile network use, especially focusing on GSM–R, and consequently FRMCS, within the framework of the methodology's development?

This paper first examines the current state of interoperability within the EU with a focus on information technology. It then identifies the main parameters for mobile applications that would be useful to include in the methodology. Among the main parameters, the current use of mobile networks in the EU is described in further detail, especially the GSM–R mobile network, with a detailed focus on the Czech Republic and its listed neighbours. In these countries, representatives of mobile applications are selected according to type and a comparison of basic parameters is made. On the basis of the findings, a discussion is initiated on the way forward to achieve relevant results. Finally, an evaluation of the research results is made.

## 2. Materials and Methods

The systematic literature research review methodology used in this paper is described by Durach, Kembro and Wieland [1]. It builds on previous research by Mulrow [2], with which it leads to transparent objective results for the evaluation of selected literature sources.

The whole process can be described in six steps:

1. Defining the areas of research intent;
2. Defining detailed target requirements;
3. Identification of appropriate scholarly sources/databases;
4. Selection of relevant literature sources;
5. Analysis of selected articles, finding connections and common conclusions;
6. Presentation of the results in quantitative and qualitative form corresponding to the selected requirements.

### 2.1. Points 1 and 2

The aim of a systematic literature search is to identify gaps in the "market of knowledge", or to find a suitable topic for which it is possible to obtain space for the creation of significant conclusions that can later lead to further practical implementation, e.g., in the form of designing an improved methodology or software development. One of the results of the research is also to gain an overview of the field of the problem being addressed.

### 2.2. Points 3 and 4

As a source of scientific sources, the generally respected Web of Science database—currently the largest and most widely used database of scientific articles of its kind—and the official database of EU documents were chosen. As of the date of this article, 15.1.2023, both databases contain approximately 100 articles devoted to the issue "Interoperability in EU with a focus on IS". Depending on the choice of the appropriate combination of keywords, the number of relevant results varies in the range of approximately 50–200. Scholarly articles and literature searches for the period 2005–2022 were taken as relevant. Two independent researchers were involved in the selection of the articles. A total of 32 sources were selected for detailed analysis.

In addition, the EU documentation database was used as a source of information. The documents are publicly displayed and available on the internet in their official form. A list of each document used is provided in the sources.

### 2.3. Points 5 and 6

First, the background of the selected sources was examined in the context of the focus of the journals and their years of publication. Keywords, references, number and quality of citations were analysed. Of course, a detailed analysis of the thematic focus of the articles, the methodologies used and the conclusions were also analysed.

The results of the systematic search show that the chosen topic—information systems and interoperability in the EU—is very relevant in the professional world. On the other hand, it should be emphasized that the chosen topic is quite broad and contains a number of subareas that would require separate research.

According to the authors of the methodology, the results of the literature review are both qualitative and quantitative.

## 3. Results

By conducting a search, the following information was obtained from the above sources.

### 3.1. Information Systems and Interoperability in the EU

According to a previous study [3], railway networks, traffic management, signalling and train safety in the countries of the current EU have had a purely national character until recently. This was due to both the historical development, when only over time the formerly separate railway networks were only connected to the pan-European network, and to strategic reasons, when the easy possibility of foreign vehicles moving on the national railway network was not desirable. It was only the huge increase in car traffic in the 1980s and especially the 1990s that led European countries to change their view of rail transport. Today, the EU is pursuing the goal of allowing trains to run unrestricted on the entire rail network of the Union and other cooperating countries, i.e., interoperability.

The main authority on railways in the EU is currently the European Railway Agency—ERA. Its objective is to contribute to the further development and effective functioning of a single European railway Area. It develops a high level of railway safety and interoperability, while improving the competitive position of the railway sector. The ERA's main objectives are described in a previous study [4]—it acts as the European authority under the 4th Railway Package issuing vehicle authorisations, single safety certificates and ERTMS (described below) trackside pre-authorisations.

ERTMS is the European Rail Traffic Management System. It is being implemented by the countries of the European Union, together with Norway and Switzerland. ERTMS project's aim is to make rail transport safer and more competitive. The reason for its introduction was described in another resource [5]—unification of the multiple interlocking systems in use will bring increased competitiveness, better interworking of freight and passenger rail services, to stimulate the European rail equipment market, reduce costs and improve the overall quality of rail transport.

ERTMS' main parts are:

- ETCS (European Train Control System)—automatic train interlocking system, based on cab signalling and spot and/or continuous track to train data transmission. It ensures trains operate safely at all times in providing safe movement authority directly to the driver through the cab display and in continuously monitoring the driver's actions;
- GSM–R—a digital radio platform for railway internal voice and data communication, providing bearer functionalities for the ETCS as the common standard control-command system (CCS), which will be described in detail below.

Further use of these systems names the resource [3]—in addition to countries in Europe, 13 countries in Asia are operating or building ERTMS (e.g., China, under the name CTCS—Chinese Train Control System). Furthermore, Turkey and Saudi Arabia

are significant ERTMS operators as well as 9 countries in Africa (e.g., Algeria, Morocco). Among the American countries, ERTMS is deployed on tracks in Brazil, Chile, Mexico and Uruguay.

The ERA ensures interoperability by issuing a technical specification for interoperability (TSI). It provides EU member states with the technical expertise to design and implement reforms. As described in a previous study [6], these TSIs define the technical and operational standards that each subsystem must meet. Relevant for this study are articles from the fields of:

Telematics applications for passenger service TSI—This TSI is described in a previous work [7] and concerns the telematics applications subsystem and applies to applications for passenger services, including systems providing information to passengers before and during the journey, reservation and payment systems, luggage management and management of connections between trains and with other modes of transport.

Telematics applications for freight service TSI—This TSI is described by authors of [8] and concerns the telematics applications subsystem and applies to applications for freight services, including information systems (real-time monitoring of freight and trains), marshalling and allocation systems, reservation, payment and invoicing systems, management of connections with other modes of transport and production of electronic accompanying documents.

The integration of the individual countries was set out in the ERTMS National Implementation Plans. CZ follows the 2017 plan. Reforms towards the implementation of the plan are continuously monitored. The status as of 2022 is provided in the supporting reforms of the CZ Technical Support Instrument 2022 [9].

The implementation of this plan is continuously monitored and published annually in status reports, along with other variables. CZ is, according to authors of [10], engaging very well with the plan.

*3.2. Mobile Applications in Rail Transport in the EU*

The mobile wireless communication system has undergone several stages of evolution over the past decades. As described in a study [11], due to the huge demand for more connections worldwide, mobile communication standards have evolved rapidly to support more users.

As the performance of mobile networks increases, mobile applications are also being developed and massively expanded and are now widely used in rail transport.

Mobile applications used in transport are linked to existing information systems or applications. Of course, the above technical specifications also apply to mobile applications to varying degrees—the main criterion is the transport sector to which the application relates—operational applications, transport planning, traveller applications, etc.

Other parameters have also been considered for mobile applications. The authors identified the following parameters as the most crucial:

- Application type (native, progressive web apps, hybrid, responsive web);
- Data network connection—mobile networks, wi-fi;
- Operating system of the mobile device where the application is running;
- Development platform including web browser support in the case of web-based technology;
- Global navigation satellite systems (GNSSs);
- Distribution of applications to users;
- Formats of transmitted data;
- Communication interfaces;
- Security.

These other parameters are also treated differently for different types of applications depending on their purpose.

- In the spirit of the drive to unify rail transport in the EU, it is therefore advisable to be as uniform as possible in the design of future mobile applications. Moreover, mobile

applications are not as robust as large-scale information systems, so it is not such a problem to adapt existing ones.

### 3.2.1. Use of Public Mobile Networks through Mobile Apps

There is currently very good coverage of quality public signals across the EU. The data collected in a study [12] show that more than 188 million EU households (97.9%) had access to at least one of the main fixed broadband access technologies in mid-2021 (excluding satellite). The Czech Republic is one of the leading countries in the European Union in terms of 4G signal coverage. The coverage of individual operators is comparable—all operators cover more than 90% of the Czech territory with 4G signals, and the coverage of 5G networks is continuously expanding.

### 3.2.2. GSM–R Mobile Network and Its Applications

As mentioned above, the GSM–R network was created for ERTMS purposes.

Since 2000, GSM–R has been used throughout Europe as a common standard for railway operations essential for interoperability, as well as in many other parts of the world.

As was described in another study [13], the limitations of GSM–R are:

- Low link rate: its maximum transmission rate per connection (9.6 kbps) is sufficient only for applications with low requirements (ETCS is around 4 kbps). Moreover, its delay, which is within the range of 400 ms, is too high for critical video applications whose acceptable delay is 100 ms;
- Interference with the GSM–R due to different sources;
- Electromagnetic (EM) transient interference due to the sliding contact between the pantograph and the catenary;
- Public GSM communications employing channels adjacent to the GSM–R frequency;
- Small system capacity: GSM–R system, with the available bandwidth 4 MHz can only support 19 channels with a 0.2 MHz bandwidth per channel, which cannot accommodate rapidly developing railway communications.

As such, it is fairly clear to all that GSM–R networks will start to expire by 2030, making their maintenance ever more costly and complex. The Union of Railways (UIC) groups, in charge of GSM–R, working hand in hand with railway operators, decided to begin looking into GSM–R's replacement.

In an analysis [14], was shown that there were two ambitions—the need to replace an obsolete technology despite its entirely satisfactory performance, and the goal of creating the enablers of digital railway operations.

The International UIC launched the first studies for a successor to GSM–R in 2012. A clear preference for 5G has emerged, a choice supported by many telecom providers and major European railway operators.

However, railway companies will have to keep their own mobile networks separate (like the current GSM–R) and will not use the public 5G mobile network because new allocated 5G bands for rail do not have enough capacity for critical data communications.

The project has been named the Future Railway Mobile Communication System (FRMCS).

According to authors of [15], led by ERA and UIC, FRMCS will be the only global rail communication standard. It is crucial that railway operators start planning the migration of their existing networks to the FRMCS standard sufficiently in advance if they are to fully use the business and technological opportunities created by them. A roadmap for this issue has been defined in Europe; it will extend until 2050, which means that GSM–R and FRM-CSFRMCS systems will operate in parallel for several years during the network migration period. Railway organisations should accelerate the complex process of modernising both their telecommunications networks and their rolling stock.

### 3.3. Implementation of EU Information Technologies Standards

In each EU country, the rules for migration into ERTMS are governed by documents [16,17], which are controlled by ERA.

### 3.3.1. GSM–R, FRMCS, ERTMS in Czech Republic

The Czech National Implementation Plan (NIP) was submitted in 2017. It is fully compliant with the ERTMS European Deployment Plan. According to the Czech NIP [18], the entire network will be equipped with ERTMS Level 2.

In Czechia, GSM–R is operated by Správa železnic, s.o. (as GSM–R CZ). Within this network, it provides nonpublic electronic communication services in accordance with EU standards, namely voice, data and fax services, and administration of subscriber SIM cards depending on the subscriber and user profile.

Currently, GSM–R is still in use in the Czechia. It is still a proven technology that is used throughout Europe and other parts of the world. It uses a common and unified spectrum and support for this network is guaranteed until at least 2030. The construction of GSM–R is still ongoing, as it is the only interoperable system that can be used according to the CCS TSI for the time being.

A working group for the migration from GSM–R to FRMCS has been set up: FRMCS Migration Scenarios (FMS). It is clear that the migration has to be gradual, similar to the GSM–R migration. According to authors of [19], at present, the detailed specifications for FRMCS have not yet been agreed and are expected to be released in 2023.

### 3.3.2. GSM–R/ERTMS in the Slovak Republic

The Slovakian NIP was delivered in 2017. It is compliant with the ERTMS European Deployment Plan with some exceptions, such as the Devinska Nova Ves-Kuty and Lanzhot-Kuty sections, which are delayed until 2030; and the Zilina node, which is also delayed (until 2023).

According to the Slovakian NIP, the entire network will be equipped with ERTMS Level 2, except for the Bratislava–Trenčín–Puchov and Zilina–Trenčín lines, which will be equipped with ERTMS Level 1 [20].

In February 2022, Railways of the Slovak Republic (ŽSR) started the trial operation of the GSM–R network on the 25 km long section Bratislava–Senec [21].

### 3.3.3. GSM–R/ERTMS in Poland

The Polish NIP was delivered in 2017. It is compliant with the ERTMS European Deployment Plan with one exception: the estimated date for ETCS operativity on the line between Lowicz and Pilawa differs from the one set in the EDP.

According to the Polish NIP, the network will have mixed ERTMS levels; some lines will be equipped with ERTMS Level 1, such as the Grodzisk Mazowiecki–Zawiercie line, and others with ERTMS Level 2, such as the Wrocław–Opole line.

Poland will not remove the Class B train protection system for the next 20 years, and the Class B radio system is expected to be removed after 2023. In fact, it is stated that there are no plans to discontinue the use of Class B train control devices; all vehicles equipped with onboard ETCS equipment must be equipped with the STM [22].

According to authors of [23], in Poland, the implementation and operation of the GSM–R network is carried out by the railway network national operator, PKP Polskie Linie Kolejowe SA. The GSM–R network is deployed on the Warszawa–Trójmiasto line. The installation of the modern ERTMS/ETCS level 2 railway traffic management system on this route has enabled the train speed to be increased to 200 km/h. As a result, travel times on the Warsaw–Gdańsk route have been reduced, with the fastest journeys taking about two and a half hours. After the installation of the GSM–R system and the ETCS upgrade, trains will also speed up on other properly prepared routes—for example, on the central railway line, they will speed up and travel at speeds of up to 250 km/h.

### 3.3.4. GSM–R/ERTMS in Hungary

The Hungarian NIP was submitted in 2018. It is not compliant with the ERTMS European Deployment Plan, because some lines have deadlines that are delayed by one or two years compared to the EDP deadlines.

According to the Hungarian NIP, the complete network will be equipped with ERTMS Level 2. Hungary plans to stop using the analogue160 Megahertz line radio system after finishing GSM–R Phase II projects (2255 km) and the Ferencváros–Kelebia–Serbian border section. In addition, it plans to use ETCS Level 2 with GSM–R on the TEN-T core network line after 2030 [24].

*3.4. Specific Mobile Applications and Their Selected Parameters*

A set of selected existing mobile applications in the Czech Republic, Slovakia, Poland and Hungary across different transport sectors was secured for the purpose of the state-of-the-art research. For these applications, the authors identified and compared key parameters, which are listed below.

ETD—Electronic Timetable Display

The ETD application is a tool for managing the tabular timetables on the mobile device of the user—the train driver. The application takes over the tabular timetables from Správa železnic, s. o.

The application is used by Czech carriers ČD and ČD Cargo.

| | |
|---|---|
| User management: | LDAP-based user directory server |
| Data transfer: | Wi-Fi, mobile network of the selected operator |
| Supported OS: | Android |
| GNSS: | GPS |
| App distribution: | Own mobile device management (MDM) |
| Developed by: | D-IS |
| Determination: | Operation of rail transport |
| Country: | CZ |

MPRIS

The MPRIS application is a mobile version of the PRIS—Operational Information System. It serves as a support for work in stations with a special focus on marshalling yards where train construction takes place. The user, directly on the track, acquires input data (train inspection, train inventory, train assignment, etc.) into the mobile device, which is then sent for processing.

The application is used by the largest Czech carrier, ČD Cargo.

| | |
|---|---|
| User management: | LDAP-based user directory server |
| Data transfer: | Wi-Fi, mobile network of the selected operator |
| Supported OS: | Android |
| GNSS: | Not used |
| App distribution: | Own MDM |
| Developed by: | ČD-IS |
| Determination: | Operation of rail transport |
| Country: | CZ |

MEKN

The MEKN app is a mobile version of the EKN—Electronic Book of Normals. It serves as a software support for the electronic familiarization of employees with documents, which aims to gradually eliminate the need of paper documents to employees.

The application is used by the largest Czech carrier, ČD Cargo.

| | |
|---|---|
| User management: | LDAP-based user directory server |
| Data transfer: | Wi-Fi, mobile network of the selected operator |
| Supported OS: | Android |
| GNSS: | Not used |
| App distribution: | Own MDM |
| Developed by: | ČD-IS |
| Determination: | Operation of rail transport |
| Country: | CZ |

### ČDCgo

The ČDCgo application is designed to monitor the status of transport of empty and loaded wagons. It is a mobile version of the Eroza portal.

The application is intended for customers of the Czech carrier, ČD Cargo.

| | |
|---|---|
| User management: | Database of ČDC portal users |
| Data transfer: | Wi-Fi, mobile network of the selected operator |
| Supported OS: | Android, iOS |
| GNSS: | Not used |
| App distribution: | Google Play, App Store |
| Developed by: | Betrian |
| Determination: | Operation of rail transport |
| Country: | CZ |

### EVAL Mobile

The EVAL mobile application is the mobile part/interface of the EVAL desktop application. The mobile application collects information about the activities performed by drivers, sends this information to EVAL agendas and processes it further. These data are the basis for the registration and planning of drivers' work and an information source for dispatchers. The LWP web interface is used to display the data.

| | |
|---|---|
| User management: | Custom storage in the Eval desktop app—by carrier |
| Data transfer: | Wi-Fi, mobile network of the selected operator |
| Supported OS: | Android |
| GNSS: | GPS |
| App distribution: | Google Play |
| Developed by: | Oltis |
| Determination: | Operation of rail transport |
| Country: | CZ, SK, HU—small carriers |

### MobiDrive Project

This project involves the development of an application for capturing operational data from railway operators. Its aim is to offer an application with lower acquisition costs and features: vehicle defects, train inventory and train documentation. The application is expected to be used by smaller carriers in Poland and Hungary.

| | |
|---|---|
| Developed by: | Oltis |
| Determination: | Operation of rail transport |
| Country: | PL, HU |

### ZSSK CARGO—Podaj

This mobile application is designed for customers of the Slovak freight carrier ZSSK CARGO. It allows the user to manage the waybills of his shipment and connect to the carrier's customer portal.

| | |
|---|---|
| User management: | The carrier's own user database |
| Data transfer: | Wi-Fi, mobile network of the selected operator |
| Supported OS: | Android |
| GNSS: | Mot used |
| App distribution: | Google Play |
| Determination: | Operation of rail transport |
| Country: | SK |

RailNavigator

RailNavigator is a platform providing services for rail freight transport. It includes a mobile application to fill a train running record, which is further linked to a web application for dispatchers.

| | |
|---|---|
| Data transfer: | Wi-Fi, mobile network of the selected operator |
| Supported OS: | Android |
| GNSS: | GPS |
| Developed by: | Rail Navigator Informatikai Fejlesztő |
| Determination: | Operation of rail transport |
| Country: | HU |

WDRflyer

WDRflyer is a car damage reporting app. It consists of a mobile app where the damage report is taken and sent, and a web portal where the received data are managed.

| | |
|---|---|
| User management: | Own user database |
| Data transfer: | Wi-Fi, mobile network of the selected operator |
| Supported OS: | Android, iOS |
| Developed by: | RailAppSolutions |
| Determination: | Operation of rail transport |
| Country: | By carrier—more EU countries |

TrainChecker

TrainChecker is an application that combines the functions of damage reporting and the creation of truck lists. The truck numbers can be photographed, other data about the trucks are retrieved from databases and some information (e.g., braking weight) can be calculated in the application.

| | |
|---|---|
| User management: | Own user database |
| Data transfer: | Wi-Fi, mobile network of the selected operator |
| Supported OS: | Android, iOS |
| Developed by: | RailAppSolutions |
| Determination: | Operation of rail transport |
| Country: | By carrier—more EU countries |

Můj Vlak

The Můj Vlak app is mobile support for travelling on ČD trains. It enables the user to search for train connections; purchase domestic tickets, seat tickets and additional ČD services; view information on the progress of the journey; monitor the current location and delay of the train; obtain an overview of closures, emergencies, changes in the train route; etc.

| User management: | Without login/DB of cd.cz portal users |
|---|---|
| Data transfer: | Wi-Fi, mobile network of the user's operator |
| Supported OS: | Android, iOS |
| GNSS: | GPS |
| App distribution: | Google Play, App Store |
| Developed by: | CHAPS |
| Determination: | Travelling public |
| Country: | CZ |

Leo Express: Stylové cestování

| User management: | Without login/DB of carrier portal users |
|---|---|
| Data transfer: | Wi-Fi, mobile network of the user's operator |
| Supported OS: | Android, iOS |
| GNSS: | GPS |
| App distribution: | Google Play, App Store, Huawei AppGallery |
| Determination: | Travelling public |
| Country: | CZ |

RegioJet

| User management: | Without login/DB of carrier portal users |
|---|---|
| Data transfer: | Wi-Fi, mobile network of the user's operator |
| Supported OS: | Android, iOS |
| GNSS: | GPS |
| App distribution: | Google Play, App Store, Huawei AppGallery |
| Determination: | Travelling public |
| Country: | CZ |

Arriva CZ

| User management: | Without login/DB of carrier portal users |
|---|---|
| Data transfer: | Wi-Fi, mobile network of the user's operator |
| Supported OS: | Android, iOS |
| GNSS: | GPS |
| App distribution: | Google Play, App Store |
| Determination: | Travelling public |
| Country: | CZ |

Vlakem na výlet

The Vlakem na výlet application, together with the vlakemnavylet.cz website, is ČD's leisure portal, which is used to search for travel destinations according to various categories.

| User management: | DB of cd.cz portal users |
|---|---|
| Data transfer: | Wi-Fi, mobile network of the user's operator |
| Supported OS: | Android, iOS |
| GNSS: | GPS |
| App distribution: | Google Play, App Store, Huawei App Gallery |
| Determination: | Travelling public |
| Country: | CZ |

IDOS

IDOS is a mobile application designed primarily for finding train, bus and public transport connections of various carriers. It provides additional services—SMS ticket, display of connections on the map, etc.

| User management: | Without login/DB of IDOS portal users |
|---|---|
| Data transfer: | Wi-Fi, mobile network of the user's operator |
| Supported OS: | Android, iOS |
| GNSS: | GPS |
| App distribution: | Google Play, App Store |
| Developed by: | CHAPS |
| Determination: | Travelling public |
| Country: | CZ |

Infotabule

The Infotabule mobile application provides rail passengers with up-to-date information from the Správa železnic, s.o. operational information systems for all stations and stops. It informs about on-line timetables, delays, platforms and tracks from which individual trains will depart.

| User management: | Without login |
|---|---|
| Data transfer: | Wi-Fi, mobile network of the user's operator |
| Supported OS: | Android, iOS |
| GNSS: | GPS |
| App distribution: | Google Play, App Store |
| Developed by: | ČD-IS |
| Determination: | Travelling public |
| Country: | CZ |

Ideme Vlakom

Ideme Vlakom is a mobile application of Železničná spoločnosť Slovensko, the national carrier, designed for travelling public. It is used to purchase travel documents, book seats, search for connections and available information about trains operated by the Railway Company of Slovakia.

| User management: | DB of ZSSK portal users |
|---|---|
| Data transfer: | Wi-Fi, mobile network of the user's operator |
| Supported OS: | Android, iOS |
| GNSS: | GPS |
| App distribution: | Google Play, App Store, Huawei App Gallery |
| Determination: | Travelling public |
| Country: | SK |

Analogously to the Ideme vlakom application, other passenger carriers in Slovakia have mobile applications with similar functions and the same parameters. A detailed list of them is not necessary for the purpose of this paper.

Cestovné poriadky

Cestovné poriadky is the Slovak equivalent of the IDOS application and has the same functionality.

| User management: | Without login/DB of CP portal users |
|---|---|
| Data transfer: | Wi-Fi, mobile network of the user's operator |
| Supported OS: | Android, iOS |
| GNSS: | GPS |
| App distribution: | Google Play, App Store |
| Developed by: | CHAPS |
| Determination: | Travelling public |
| Country: | Slovensko |

Zpoždění vlaků SK

Zpoždění vlaků SK app displays, on a map or as a list, all trains currently running on the Slovak rail network. Trains can be managed in favourites, filtered according to various criteria, etc. The data source is ŽSR.

| | |
|---|---|
| User management: | Without login |
| Data transfer: | Wi-Fi, mobile network of the user's operator |
| Supported OS: | Android |
| GNSS: | GPS |
| App distribution: | Google Play |
| Developed by: | Marek Scholtz |
| Determination: | Travelling public |
| Country: | SK |

Rozklad-PKP

Rozklad-PKP is a mobile app for planning your train journey. It includes timetables and other information for all passenger rail transport providers in Poland.

| | |
|---|---|
| User management: | Without login |
| Data transfer: | Wi-Fi, mobile network of the user's operator |
| Supported OS: | Android, iOS |
| GNSS: | GPS |
| App distribution: | Google Play, App Store |
| Determination: | Travelling public |
| Country: | PL |

BILKOM Bilety

BILKOM is a Polish application for purchasing and managing tickets and other additional services. It includes rail connections offered by PKP Intercity, PolRegio, PKP SKM, Łódź Agglomeration Railway, Wielkopolskie Railways, Malopolska Railways, Arriva RP and Koleje Śląskie.

| | |
|---|---|
| User management: | Without login/DB of Bilkom.pl portal users |
| Data transfer: | Wi-Fi, mobile network of the user's operator |
| Supported OS: | Android, iOS |
| GNSS: | GPS |
| App distribution: | Google Play, App Store |
| Determination: | Travelling public |
| Country: | PL |

Opóźnienia pociągów

The app shows the train delay at a given station in Poland. The functions of the application are station search, list of departing and arriving trains in the station, list of stations on the route of the train, current position of the train in each station, information about the carrier, notifications, etc.

| | |
|---|---|
| User management: | Without login |
| Data transfer: | Wi-Fi, mobile network of the user's operator |
| Supported OS: | Android |
| GNSS: | GPS |
| App distribution: | Google Play |
| Determination: | Travelling public |
| Country: | PL |

MÁV-START

The MÁV-START app is used to plan journeys on passenger trains in Hungary. It has functions to find connections, buy tickets, station information, etc.

| | |
|---|---|
| User management: | Without login/DB of MÁV portal users |
| Data transfer: | Wi-Fi, mobile network of the user's operator |
| Supported OS: | Android, iOS |
| GNSS: | GPS |
| App distribution: | Google Play, App Store, Huawei App Gallery |
| Determination: | Travelling public |
| Country: | HU |

The key application parameters have been clearly listed and categorised by application type in Table 1:

**Table 1.** Key parameters for the main types of mobile applications.

| Parameter | Operational Apps | Apps for Public—General | Apps for Public—Specific Carrier |
|---|---|---|---|
| Data transfer | Public mobile network, operator by carrier Wi-Fi | Public mobile network, user's operator Wi-Fi | Public mobile network, user's operator Wi-Fi |
| App distribution | Own MDM | Public online distribution service | Public online distribution service |
| GNSS | GPS (if used by the application) | GPS (if used by the application) | GPS (if used by the application) |
| Operation system | Android, iOS | Android, iOS | Android, iOS |
| User management | Carrier/operator's own storage | Storage is handled by the application /without login | Storage is handled by the application /without login |

The comparison of parameters shows that some parameters differ depending on the application. In particular, the distribution on the device is different. For applications that are intended to support transport operations and are therefore used on the service devices of the carrier/operator, the distribution to the device is mostly realized by the MDM itself. Another difference is user management—for operational applications, users are usually stored and managed in an internal structure (e.g., in the environment of the Czech Railways group, this is a directory server operated on the basis of the LDAP protocol). For public applications, it is either possible to use the application without logging in, or users are stored in a repository linked to the application. It was found that it does not matter whether the app is for travel in general or only for one specific carrier.

On the other hand, some parameters are the same—the use of GNSS (if used by the application)—for the applications studied, only the GPS system was found to be used. The applications use only public mobile networks for data transmission—no application uses the GSM–R network. For operational applications on service equipment, the operator is chosen by the application owner; for public applications the network used is determined by the specific user of the device. Of course, all of the above applications can also be connected to the data network via Wi-Fi.

## 4. Discussion

This article summarised the basic EU interoperability rules for the creation of a single European railway area, focusing mainly on telematics applications. After assessing their implementation in the form of NIPs for the Czech Republic, Slovakia, Poland and Hungary, it was concluded that these countries are more or less following the set plans and it is therefore possible to consider applying a uniform follow-up procedure, also in the field of mobile transport applications.

The key parameters of mobile applications were identified. As the first of the key parameters, mobile networks were selected for detailed analysis in this study as a means of connecting mobile applications to data or voice services. Thus, the current state of mobile networks in the EU in general and in the Czech Republic in detail was assessed, with a particular focus on GSM–R. For this network, whether the network could be used for the operation of mobile transport applications was examined, and it was found that in the current state of the network and the conditions of the operators, the network is not

ready for more or less massive operation of mobile transport applications. The successor of GSM–R, the FRMCS network, was also mentioned and its actual status, which corresponds with the preparation of specifications, was described.

In order to evaluate the current state of the art for real mobile applications from different points of view, a set of sample transport applications in listed countries was selected. The values of key parameters were determined for these applications. For the mentioned selected parameter—mobile networks—it was found that in fact no real used mobile transport application uses GSM–R network for data or voice transmission. The mobile networks used are highly dependent on the type of mobile application.

The findings have led to the following conclusions: as the implementation of the Single European Railway Area and interoperability is on schedule, it makes sense to address a common design methodology for transport mobile applications in the context of interoperability in order to harmonise their design and operation and thus make their development and deployment more efficient across the EU. The assessment of the current state of the GSM–R network has led to the abandonment of the original idea of the possibility of using GSM–R for mobile transport applications. In order to consider the use of the FRMCS network, it is necessary to wait for more precise specifications to be issued in 2023. The assessment of the actual status of the selected applications has answered (or added to) the original question of how far it will be possible to unify the design of mobile applications—namely that different groups of transport applications (operational, public passenger, etc.) place different demands on key parameters. This means that these groups need to be approached individually when designing the methodology.

The results of the research and the following considerations are important both for the identification of key parameters whose analysis and recommendations for their unification will be of real importance in terms of the integration of mobile transport applications into the EU interoperability package and, more importantly, for the possibility of applying these recommendations in real terms with regard to the development, operation and use of applications. The original assumptions were re-evaluated by the study, namely the limited use of the GSM–R network and the need for more approaches for different types of mobile transport applications.

Unfortunately, it was not possible to compare the findings with other studies because no study with a similar focus has been published yet.

The limitation of this study lies in its scope, as it has only been carried out for one key parameter—mobile networks. Thus, only a part of the problem has been examined. There is also no consideration of the extent to which the proposals could be applied in the sophisticated challenging environment of rail transport in the EU, where other than technical influences—economic, political, social, etc.—are in game.

Therefore, for further work on the design of a broader methodology, it will be appropriate to complement the study with follow-up analyses of other key parameters. An appropriate complement would be to analyse the possibilities of applying the proposed methodology in practice, with specific carriers, developers, etc.

As a next part of the work, it is also advisable to further monitor the next development of the FRMCS network's specifications and to adapt the proposals accordingly. The result of the complete study should be the design of a mobile application according to the proposed methodology, which will verify the original assumptions, the results obtained and the proposals made.

## 5. Conclusions

In conclusion, after a search of EU documents issued to support interoperability and the current state of their implementation, it is possible to find uniform recommendations for mobile applications. Concerning the parameter studied—mobile networks—the coverage and quality situation of public mobile networks in the EU is very good. After analysing the current state of the GSM–R network, it was concluded that its use for mobile transport applications is not currently beneficial. Recommendations on the choice of the mobile

network will mainly depend on the type of mobile transport application (operational applications are usually linked to company contracts with the operator; for the public, the choice of the network is a matter for the specific user of the application). As a follow-up work, it is advisable to perform the same analysis for other key parameters of mobile applications, which will form the basis for the design of a unified methodology for the design of mobile transport applications. It is also advisable to conduct a study of the possibilities and willingness to apply the proposed methodology in the development of mobile applications in practice, taking into account the economic, political, etc., situation in the EU.

**Author Contributions:** Conceptualization, P.Š.; Methodology, S.M.; Validation, S.M.; Formal analysis, P.Š. and S.M.; Investigation, P.Š.; Resources, P.Š. and S.M.; Data curation, S.M.; Writing—original draft, P.Š. and S.M.; Writing—review and editing, P.Š. and S.M.; Supervision, S.M.; Project administration, P.Š.; Funding acquisition, P.Š. All authors have read and agreed to the published version of the manuscript.

**Funding:** This research received no external funding.

**Data Availability Statement:** The data presented in this study are available within the article.

**Conflicts of Interest:** The authors declare no conflict of interest.

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
