# Peer review of "A Background to a Methodology for Transport Mobile Applications Designing with a Focus on EU Interoperability"

_infrastructures, doi:10.3390/infrastructures8020034_

Round 1

Reviewer 1 Report

Regarding the content, I do not have any changes to recommend, it makes a good literary review to support the relevance of the problem to be studied and a good structuring of the content, it uses the correct methodology for this type of study and it is a consistent and well-detailed methodology to give significance to the results they show, makes a good discussion of the results, and marks the conclusion obtained well.

Although I advise looking at these things:

On line 633, table 2 should be table 1.

A table or figure should never appear without reference to it in the previous text.

In the section “1. Introduction”, it is necessary to develop a deeper analysis of the justification, motivation and structure of the article.

In the section “5. Conclusions”, it is necessary to develop a deeper analysis of the conclusions and implications of the study. In addition to the possible future lines of research opened with this research.

And the references in the 'References' section must follow the model set by the journal. You must correct the errors that exist. Look at this in the template.

Author Response

Dear reviewer, thank you for your expert feedback on the article and for the recommendations. The article has been revised according to the recommendations and the responses are attached.

Reviewer 2 Report

Dear authors,

Upon examination of this article, it can be concluded that this paper focuses on an important task in rail operation concerning interoperability and mobile applications. Although the goal of the paper is clearly defined and the article is written systematically, it is too theoretical. The title does not fully describe the content of the article.

Obviously, the article outputs have limitations:

ü The authors presented results of their research but in this form, it does not meet the conditions to be published in this journal.

ü Section 3.3. “Implementation of EU information technologies standard” is taken from known sources – more than 7 pages just a theoretical characteristic of well-known systems.

ü The chapter “Introduction” needs to be significantly expanded and reworked.

ü In the same way the chapters “Discussion” and “Conclusion” must be expanded.

ü It is not clear what the authors' contribution is in practical terms. Try to explain the benefits of your research in more detail.

ü In the same way, the cited sources of literature are not listed in the text, so it is not clear whether this is a claim of the authors or another source.

Thank you very much and I wish you all the best in your efforts.

Reviewer

Author Response

(The authors gave the same response as above.)

Round 2

Reviewer 2 Report

Dear authors, Thank you for revising your manuscript, I recommend its publication.